# ⊞ MATHDIAL: A Dialogue Tutoring Dataset with Rich Pedagogical Properties Grounded in Math Reasoning Problems

**Jakub Macina**\*🚀⛰️    **Nico Daheim**\*🏛️    **Sankalan Pal Chowdhury**\*⛰️
**Tanmay Sinha**🌳    **Manu Kapur**⬥    **Iryna Gurevych**🏛️    **Mrinmaya Sachan**⛰️

🚀ETH AI Center    ⛰️Department of Computer Science, ETH Zurich
🏛️Ubiquitous Knowledge Processing Lab (UKP Lab), Department of Computer Science
and Hessian Center for AI (hessian.AI), TU Darmstadt
🌳National Institute of Education, Nanyang Technological University
⬥Professorship for Learning Sciences and Higher Education, ETH Zurich
jakub.macina@ai.ethz.ch

## Abstract

While automatic dialogue tutors hold great potential in making education personalized and more accessible, research on such systems has been hampered by a lack of sufficiently large and high-quality datasets. Collecting such datasets remains challenging, as recording tutoring sessions raises privacy concerns and crowdsourcing leads to insufficient data quality. To address this, we propose a framework to generate such dialogues by pairing human teachers with a Large Language Model (LLM) prompted to represent common student errors. We describe how we use this framework to collect ⊞ MATHDIAL , a dataset of 3k one-to-one teacher-student tutoring dialogues grounded in multi-step math reasoning problems. While models like GPT-3 are good problem solvers, they fail at tutoring because they generate factually incorrect feedback or are prone to revealing solutions to students too early. To overcome this, we let teachers provide learning opportunities to students by guiding them using various scaffolding questions according to a taxonomy of teacher moves. We demonstrate ⊞ MATH-DIAL and its extensive annotations can be used to finetune models to be more effective tutors (and not just solvers). We confirm this by automatic and human evaluation, notably in an interactive setting that measures the trade-off between student solving success and telling solutions. The dataset is released publicly.

○ https://github.com/eth-nlped/mathdial

## 1 Introduction

Dialogue tutoring systems have demonstrated significant potential in augmenting learning outcomes across various domains (Wollny et al., 2021; Ji

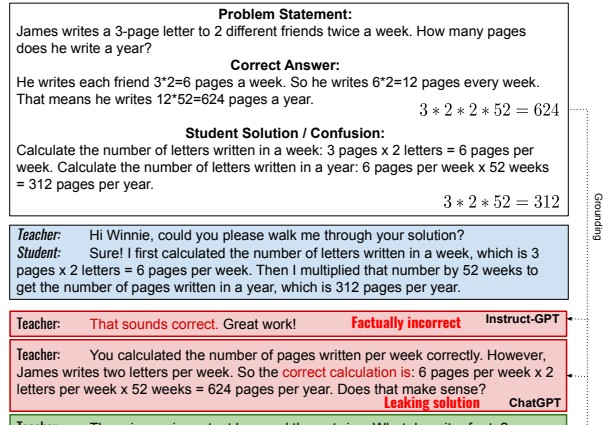

Figure 1: Current models achieve high accuracy in solving MWPs but struggle with teaching since they often give incorrect feedback or reveal directly the solution too early. ⊞ MATHDIAL mitigates this using scaffolding questions and grounding annotations.

et al., 2023). However, the progress of scaling them is considerably hindered by a lack of high-quality datasets, which actually provide students with space for exploration by scaffolding their learning (Tack and Piech, 2022; Macina et al., 2023). The current datasets are frequently marred with issues like low pedagogical quality, are too small, or focus on noisy classroom settings. While recording tutoring sessions might be a scalable alternative, it bears strong privacy concerns (Demszky and Hill, 2023). On the other hand, crowdsourcing dialogues is costly, requires synchronizing annotators, and can lead to insufficient quality due to poor annotator training (Stasaski et al., 2020).

At the same time, recent advancements in Large Language Models (LLMs) have enabled significant improvements in generative dialogue systems (Budzianowski and Vulić, 2019; Thoppilan et al., 2022; Xu et al., 2023) and simultaneously shown

---

\*Equal contribution.

| Dataset | Domain | Dialogues | Dialogic Pairs | Settings | Grounding Information | Teacher Moves | Bigram Entropy | Uptake | Avg. words per utterance |
|---|---|---|---|---|---|---|---|---|---|
| **MATHDIAL** (ours) | Math | 2 861 | 14 197 | 1:1 semi-synthetic | confusion, answers | 4 | 3.54 | 0.83 | 17.3 |
| CIMA (Stasaski et al., 2020) | Language | 391 | 3 315 | 1:1 role-playing | image, answer | 5 | 3.12 | 0.83 | 13.0 |
| TSCC (Caines et al., 2020) | Language | 102 | 2 013 | 1:1 tutoring | ✗ | 5 | 3.55 | 0.66 | 12.3 |
| TalkMoves (Suresh et al., 2022) | Science | 567 | 9 280 | classroom | ✗ | 10 | 2.93 | 0.67 | 9.6 |
| NCTE (Demszky and Hill, 2023) | Math | 1 660 | 2 348 | classroom | ✗ | ✗ | 3.57 | 0.76 | 29.2 |

Table 1: Comparison of dialogue tutoring datasets. ▦ MATHDIAL has grounding annotations, and is significantly larger while keeping high diversity and utterance lengths.

great success in reasoning over educational domains, such as math problems (Cobbe et al., 2021; Wei et al., 2022; Wang et al., 2023b; OpenAI, 2023). However, this has not yet translated to improvements in dialogue tutoring systems, as showcased by the lack of pedagogical understanding and factually incorrect behaviour of GPT-3 (Tack and Piech, 2022) and open-source LLMs (Macina et al., 2023). Figure 1 shows examples of generations that reveal information to students too early and misunderstand their solutions. This is also confirmed in our human evaluation: when asked ChatGPT to tutor a student as a teacher, it directly **reveals the solution 66% of times** and **provides incorrect feedback 59% of times** (cf. Section 6.3).

To address these issues, we collect and present a dialogue tutoring dataset called ▦ MATHDIAL . The dataset has rich tutoring quality which we measure by **equitable tutoring** (Tanner, 2013): providing opportunities for the student to learn, think and explore potential solutions. For this, we take inspiration from human tutoring strategies (Nye et al., 2014) and active learning approaches in classrooms (Freeman et al., 2014) that show a positive impact on student learning gains.

We collect our dataset using a novel data collection approach. This approach pairs human teachers with an LLM that simulates students and their errors, which the same teachers rate as representative of real students in our study. ▦ MATH-DIAL is grounded in math word problems and student confusions and therefore provides a challenging testbed for creating faithful and equitable dialogue tutoring models that can reason over complex data. Figure 1 shows one dialogue from ▦ MATH-DIAL , where a teacher scaffolds student learning by asking an interactive scaffolding question instead of leaking the solution.

We benchmark various models on the task of generating tutor responses for ▦ MATHDIAL , using both finetuning and prompting. We find that finetuning smaller open-source LLMs on our dataset can make them significantly more equitable and faithful to the teaching material than prompting larger LLMs (Section 6.3). Moreover, we propose an interactive, end-to-end tutoring simulation between a teacher and student model where we measure a trade-off between student solving success and teachers directly revealing answers in (Section 6.4). **Open-source LLMs that are finetuned on our dataset achieve similar student-solving success as ChatGPT while telling solutions less often.** Finally, we highlight open challenges on this dataset, such as generalization to new problems.

## 2 Background & Related Work

### 2.1 Dialogue Datasets & Collection Methodologies

Research on task-oriented dialogue systems has mainly focused on customer service, for instance, restaurant reservations (Henderson et al., 2014; Gašić et al., 2014). Notably, Wen et al. (2017) collect such dialogues with the Wizard-of-Oz (WoZ) paradigm (Kelley, 1984), where crowdworkers are connected to roleplay interlocutors. One plays the user who interacts with the system, and the other roleplays the system and is often exclusively given access to domain knowledge. WoZ has been used to collect many popular datasets, such as Multi-WoZ (Budzianowski et al., 2018) and extensions (Kim et al., 2020; Zhu et al., 2020), Taskmaster (Byrne et al., 2019), and open-domain datasets like Wizard-of-Wikipedia (Dinan et al., 2019). Other collection methods include crowdworkers filling dialogue outlines (Shah et al., 2018; Rastogi et al., 2020; Majewska et al., 2023), or scraping from the web (Li et al., 2017; Dziri et al., 2019).

Multiple works have shown shortcomings in using non-expert crowdworkers. For instance, document-grounded corpora often contain hallucinations in ground-truth data (Dziri et al., 2022), and task-oriented corpora tend to suffer from annotation errors and low lexical diversity (Casanueva et al., 2022). More closely related to this work, current tutoring corpora lack sufficient tutoring quality

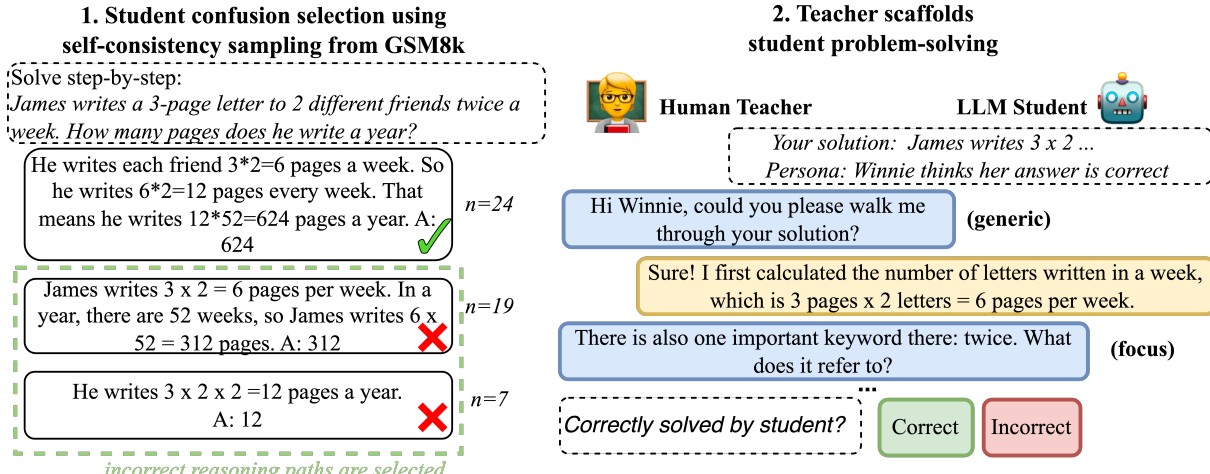

**1. Student confusion selection using self-consistency sampling from GSM8k**

Solve step-by-step:
*James writes a 3-page letter to 2 different friends twice a week. How many pages does he write a year?*

He writes each friend 3*2=6 pages a week. So he writes 6*2=12 pages every week. That means he writes 12*52=624 pages a year. A: 624 ✓ — *n=24*

James writes 3 x 2 = 6 pages per week. In a year, there are 52 weeks, so James writes 6 x 52 = 312 pages. A: 312 ✗ — *n=19*

He writes 3 x 2 x 2 =12 pages a year. A: 12 ✗ — *n=7*

*incorrect reasoning paths are selected*

**2. Teacher scaffolds student problem-solving**

Human Teacher      LLM Student

*Your solution: James writes 3 x 2 ...*
*Persona: Winnie thinks her answer is correct*

Hi Winnie, could you please walk me through your solution? **(generic)**

Sure! I first calculated the number of letters written in a week, which is 3 pages x 2 letters = 6 pages per week.

There is also one important keyword there: twice. What does it refer to? **(focus)**

*Correctly solved by student?*   Correct   Incorrect

Figure 2: Overview of the data collection pipeline: First, student confusions are oversampled from an LLM and sorted by frequency. Then, a human teacher synchronously interacts with a student simulated by an LLM that is instructed with a student profile and incorrect solution.

(Tack and Piech, 2022; Macina et al., 2023).

MATHDIAL mitigates these issues by adapting the WoZ paradigm to using human teachers as experts in collaboration with an LLM.

## 2.2 Dialogue Tutoring Corpora & Teacher Moves

Theoretical and empirical studies have shown the importance of *questioning* in human learning (Roscoe and Chi, 2008; Shahriar and Matsuda, 2021; Shridhar et al., 2022). Therefore, prior research has explored which types of questions in tutoring conversations improve student learning. Nye et al. (2014), for instance, show the effectiveness of deep reasoning questions, and (Howe et al., 2019) find that elaboration and challenging of previous contributions can benefit student learning. This has led to a series of human-authored dialogue tutoring systems, like AutoTutor (Nye et al., 2014), which guide students in problem-solving using natural language explanations. Assisting students to succeed in complex tasks commonly referred to as scaffolding (Reiser, 2004; Anghileri, 2006). More recently, several rule-based dialogue systems with predefined goals have been proposed (Ruan et al., 2019; Winkler et al., 2020; Cai et al., 2021), but scaling them requires extensive human authoring and quickly becomes complex. As a consequence, building effective automatic tutors at scale remains an open problem.

While data-driven approaches seem like a promising direction (Macina et al., 2023; Wang et al., 2023a), only a limited number of tutoring corpora are publicly available to our knowledge: CIMA (Stasaski et al., 2020), TSCC (Caines et al., 2020), TalkMoves (Suresh et al., 2022), and NCTE (Demszky and Hill, 2023). All of them suffer from several limitations, such as missing grounding information (TSCC, TalkMoves, NCTE), low tutoring quality (CIMA), small dataset sizes (all), or a focus on noisy classroom scenarios (see Table 1).

## 2.3 Synthetic Dialogue Data Creation

LLMs have recently found their way as synthetic dialogue dataset generators due to their increasingly human-like behaviour. Both methods using fine-tuning (Dai et al., 2022) and prompting (Kim et al., 2022; Chen et al., 2023) haven been proposed. The human-like behaviour also manifests in them showing similar biases in logical reasoning as humans (Dasgupta et al., 2022; Binz and Schulz, 2023), and can be comparable to gold-human annotations for generation tasks (Ziems et al., 2023). Consequently, they have been used to simulate students for teacher training (Markel et al., 2023), suggesting that one might also rely upon them to create meaningful tutors. However, Tack and Piech (2022); Macina et al. (2023) show that they can not yet perform well as teachers out-of-the-box, because they often incorrectly assess student solutions and reveal answers too quickly.

## 3 ▦ MATHDIAL Collection Pipeline

This section introduces a framework for collecting high-quality tutoring conversations, highlighted in Figure 2. The core idea behind it is to connect

| Category | Intent | Example |
|---|---|---|
| **Focus** | Seek Strategy | So what should you do next? |
| | Guiding Student Focus | Can you calculate . . . ? |
| | Recall Relevant Information | Can you reread the question and tell me what is . . . ? |
| **Probing** | Asking for Explanation | Why do you think you need to add these numbers? |
| | Seeking Self Correction | Are you sure you need to add here? |
| | Perturbing the Question | How would things change if they had . . . items instead? |
| | Seeking World Knowledge | How do you calculate the perimeter of a square? |
| **Telling** | Revealing Strategy | You need to add . . . to . . . to get your answer. |
| | Revealing Answer | No, he had . . . items. |
| **Generic** | Greeting/Fairwell | Hi . . . , how are you doing with the word problem? |
| | | Good Job! Is there anything else I can help with? |
| | General inquiry | Can you go walk me through your solution? |

Table 2: Teacher moves with examples of utterances and their intents from the 📊 MATHDIAL dataset.

an expert annotator, who roleplays a teacher, with an LLM that simulates the student.[1] We use this methodology to collect dialogues based on GSM8k (Cobbe et al., 2021), a diverse collection of grade school multi-step math word problems (MWPs).

First, we estimate student confusion for a given MWP by using temperature sampling to obtain diverse solutions from an LLM. We then select the most frequent incorrect solution. Therefore, each tutoring dialogue deals with the solution of exactly one MWP and one confusion. As a next step, we pair a human teacher with the LLM to create a dialogue that should resolve the confusion. We ground the LLM in one of six student profiles. These student profiles consist of common misconceptions of students learning algebra, such as struggling to recognize the problem type, and are taken from Booth et al. (2017). A detailed description of these profiles is found in Section C.

The teacher has access to the MWP and its correct step-by-step solution, as well as the initial student confusion (cf. Figure 7). Then, the teacher is tasked to guide the student to solve the problem by employing a sequence of scaffolding moves, which we refer to as a teaching strategy. The teachers themselves can use their expertise to determine the strategy but are required to select the current move before writing a response, as we have found this to lead to more diverse pedagogical patterns. We describe these moves in Section 3.4. The dialogue ends when the teacher marks the problem as solved or a certain time limit is reached.

In addition to the collected dialogues, we obtain metadata that future work can explore for building more effective tutor models. In particular, for each dialogue 📊 MATHDIAL contains the MWP,

step-by-step solution, the exact step that led to student confusion, and annotations indicating if it was resolved over the course of the dialogue. Step-by-step and student solutions are also provided as equations.

### 3.1 Teacher Selection

We recruit professionals with teaching experience through Prolific[2]. We only select teachers who have completed at least 500 submissions and achieved a 100% completion rate. Annotators read guidelines for the task in an initial training phase (cf. Section D.3) and then complete a test on an example conversation to assess their understanding of the task. We only select annotators with 100% test scores for further rounds of data collection, similar to Zhang et al. (2023). We employ 91 expert annotators, of which 71 identify as female and 18 as male. The majority of annotators are nationals of the UK, followed by the USA, Canada, Australia, India, and Germany, with a median age of 39 years.

### 3.2 Problem & Confusion Selection

We employ an LLM to generate plausible student confusions and base the dialogues on them. We pick the most frequent incorrect solution sampled from ChatGPT (gpt-3.5-turbo) (Ouyang et al., 2022) using chain-of-thought prompting. To be precise, we first use temperature sampling to obtain $N = 50$ reasoning paths for every MWP in GSM8k, with $T = 0.7$ and no top-k truncation Wang et al. (2023b). Then, we group incorrect solutions according to their final numeric answer and pick one from the set with the largest cardinality. More details can be found in Appendix B. As we will show in Section 4.1, teachers think that the

---

[1]In contrast, in WoZ two users are connected, with one simulating a system.

[2]https://www.prolific.co

majority of sampled confusions are plausible and could also have been made by a real student.

### 3.3 Student Turn Generation

We use InstructGPT (text-davinci-003) (Ouyang et al., 2022) to generate student turns. We prompt the model with the previous dialogue history and additional information that grounds the next turn. The prompt contains the MWP, the initial student confusion, as well as the student profile which explains the type of confusion and persona of the student.

### 3.4 Taxonomy of Teacher Moves

This section defines the taxonomy of all teacher moves that are used in ▦ MATHDIAL . We base the first two on the work of Reiser (2004), who suggest that scaffolding strategies can be split into two main categories: structure and problematize. These form the basis for the Focus and Probing moves employed in our study. Focus is used to constrain the student to make direct progress towards solving the problem. Probing is used to generalize certain aspects of the problem which allows the student to explore its underlying concepts. More concretely, a teacher might construct a new, related problem that targets only one specific concept that is needed to solve the original MWP. However, scaffolding might also fail, for example when a student gets stuck. Then, teachers may need to reveal parts of the answer. This is called Telling. Finally, turns that just serve as conversational elements and have limited pedagogical value are classed as Generic. Table 2 lists finer-grained intents for each of these four categories along with a set of accompanying examples.

## 4 ▦ MATHDIAL Analysis

We quantitatively evaluate the collected tutoring dialogues to assess their quality. For this, we outline descriptive statistics in Table 1. First of all, we can see that our dataset is significantly larger in terms of the number of dialogues and utterances than all related datasets that are listed. By open-sourcing such a large dataset, we fill a crucial gap of sufficiently-sized open-source tutoring corpora which has so far hindered research in the area (Macina et al., 2023).

Furthermore, ▦ MATHDIAL exhibits a higher diversity, measured in bigram entropy (Zhang et al., 2018), than CIMA and TalkMoves. The diversity

is similar to NCTE and TSCC which consist of transcripts of classroom and one-to-one tutoring sessions, respectively. This supports the observation that expert annotators tend to create more diverse utterances than untrained crowdworkers (Casanueva et al., 2022), and also that LLMs can be used to generate diverse tutoring dialogues. Finally, we measure the Uptake (Demszky et al., 2021) of annotated teacher utterances. Uptake indicates how coherent the teacher's utterance is with respect to the previous student's turn. We find that ▦ MATH-DIAL and CIMA have similar uptake. Both surpass the other datasets in our comparison.

### 4.1 How well can LLMs simulate students?

Our collection methodology relies on LLMs for simulating students. Therefore, it is crucial to ensure that the turns simulated by the LLM also match what a teacher would expect of a real student, who in our case is a sixth grader. In this section, we evaluate this quantitatively.

Figure 3 shows that annotators rate the majority of generations by the model positively along two dimensions. The first one says that the confusion of the student is typical confusion of a sixth grader. The second one says that the interaction with the student as a whole is as expected of a sixth grader. We release these annotations with our final dataset which allows users of ▦ MATHDIAL to filter out utterances that are of a lower quality.

Moreover, LLMs can be prone to incorrect arithmetic calculations. Therefore, we asked annotators to distinguish conceptual errors from such simple calculation mistakes. Arithmetic errors may be easily resolved through calculators but conceptual errors are likely to require tutors to resolve them, for example by scaffolding. Annotators identified around $80\%$ of the confusions as conceptual, leaving around a fifth containing arithmetic errors. Again, we include these annotations to allow for data filtering.

### 4.2 Which teaching strategies do annotators choose?

In this Section, we evaluate when teachers use which teacher moves in the conversations. Figure 4 shows that teachers most frequently use Focus questions which are found in $37\%$ of utterances. Focus is followed by Generic and Probing. Telling is the rarest move. To validate these annotations, we sampled 17 conversations consisting of 102 teacher utterances and asked two independent annotators to

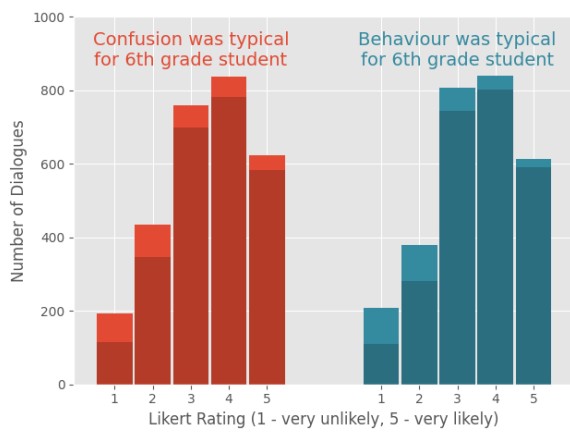

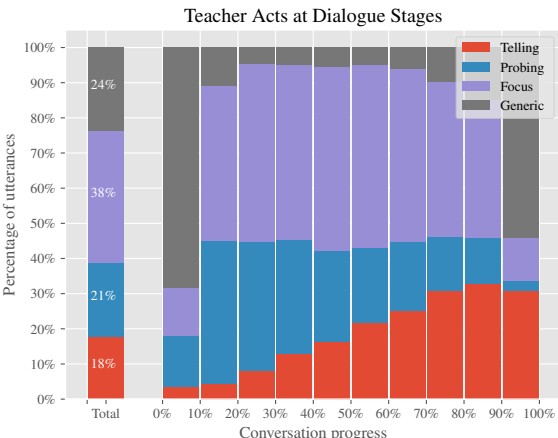

Figure 3: Teacher judgments on the ability of Instruct-GPT to simulate students. Teachers rate the simulated behaviour as largely plausible. Lighter regions on top account for questions where the confusion was not resolved.

Figure 4: Overall distribution of teacher moves (left) and their distribution at each dialogue step (right). Teachers tend to start with Focus and Probing and then increasingly use Telling as the conversation progresses.

annotate their moves. We obtain an agreement of $\kappa = 0.60$ between the two annotators and $\kappa = 0.49$ and $\kappa = 0.34$, respectively, between either of the annotators and the teacher. We note that Probing and Focus appear to be particularly challenging to distinguish and acknowledge that the boundary between them may be subjective. Merging these two categories into one larger 'scaffolding' category improves agreements to $\kappa = 0.67$, $\kappa = 0.75$ and $\kappa = 0.55$. Our observations are in line with related works that have shown low inter-annotator agreement between experts for detailed teacher moves in classroom settings (Kelly et al., 2020).

The sequence of moves employed by the teachers constitutes their teaching strategy which we analyze in the following. Figure 4 shows the distribution of teacher moves for different stages of the conversations. We find that the initial utterance by the teacher is usually generic and serves as a conversation opener, oftentimes by asking the student to repeat the question or solution attempt. During the conversation, teachers mainly use scaffolding to either probe the student or focus the conversation on a specific part of the problem. The more the conversations progress the more likely teachers are to resort to Telling because students often get stuck at a specific subproblem and are unable to resolve it themselves. As a consequence, less Probing is used. This has been shown to keep students engaged in the conversation who otherwise become frustrated by being stuck (VanLehn, 2011).

### 4.3 How often can student confusion be resolved?

The goal of ▤ MATHDIAL is to enable building tutors that can help students resolve their confusion. Therefore, we would like to know how often teachers can do so in our collected data. This is annotated by the teachers themselves, who assessed that they were successful in almost 89% of the conversations. In ca. 75% of the conversations by using mainly scaffolding questions, and only in around 14% by revealing the majority of the answer. The conversations in which confusions could not be resolved can still be useful, as they, for instance, can be used to train classifiers to determine when human intervention in such tutoring sessions is required.

## 5 Modeling Tutors with ▤ MATHDIAL

We focus our initial studies on ▤ MATHDIAL on the task of tutor response generation. *Tutor response generation* aims to model the teacher in a dialogue by generating follow-up turns to guide the student towards learning and solving the problem. In the following subsections, we compare different finetuned and prompted language models on the task and evaluate how much detailed information that can be given to the model, such as step-by-step solutions of the MWP, influence performance.

### 5.1 Training details

We use neural conditional language models that given a tutoring dialogue history $u_1^T$, grounding

| Model | MATHDIAL | | | | | MATHDIAL$_{seen}$ | | MATHDIAL$_{unseen}$ | |
|---|---|---|---|---|---|---|---|---|---|
| | sBLEU ($\uparrow$) | BERTScore ($\uparrow$) | KF1 ($\uparrow$) | BERTScore ($\uparrow$) | Uptake ($\uparrow$) | sBLEU ($\uparrow$) | KF1 ($\uparrow$) | sBLEU ($\uparrow$) | KF1 ($\uparrow$) |
| | ($u_{T+1}, \hat{u}_{T+1}$) | | ($u_{T+1}$, MWP) | | ($u_T, u_{T+1}$) | | | | |
| BART$_{139M}$ | 4.5 | 52.0 | 15.0 | 46.3 | 86.6 | 5.2 | 16.0 | 3.3 | 13.5 |
| BART$_{406M}$ | 5.7 | 52.8 | 16.1 | 47.2 | 87.9 | 6.5 | 16.1 | 4.3 | 16.1 |
| T5$_{250M}$ | 7.2 | 51.1 | **27.2** | **54.3** | **94.8** | 8.4 | **27.7** | 5.2 | **26.3** |
| T5$_{780M}$ | 9.0 | 53.8 | 23.0 | 51.4 | 91.6 | 10.7 | 23.1 | 6.3 | 23.7 |
| Flan-T5$_{250M}$ | 8.2 | 52.9 | 23.5 | 52.0 | 92.0 | 9.6 | 24.2 | 5.7 | 22.4 |
| Flan-T5$_{780M}$ | **9.7** | **55.0** | 22.1 | 51.5 | 91.7 | **11.3** | 22.5 | **6.9** | 21.6 |
| Flan-T5$_{3B}$ | 7.8 | 54.8 | 17.7 | 48.4 | 88.8 | 8.9 | 18.1 | 5.9 | 17.0 |
| OPT$_{125M}$ | 3.9 | 51.9 | 12.3 | 44.4 | 81.7 | 4.4 | 13.1 | 3.4 | 11.2 |
| OPT$_{1.3B}$ | 3.8 | 52.1 | 11.5 | 44.2 | 82.6 | 4.4 | 12.5 | 2.9 | 9.9 |
| ChatGPT (0-shot) | 2.2 | 47.7 | 22.6 | 50.3 | 92.7 | 2.1 | 22.8 | 2.3 | 22.2 |

Table 3: Results of finetuned and zero-shot prompted models on the tutor response generation task. We find that i) models finetuned on our dataset can outperform much larger prompted models, ii) there is still a gap in terms of generalization, iii) simply scaling the same pretrained model does not immediately improve results.

information $\mathcal{K}$, and a teacher move $\mathcal{A}$, we wish to generate a continuation of the dialogue $u_{T+1} \subset \mathcal{V}^*$. Here $\mathcal{V}^*$ denotes all strings that can be constructed from the model vocabulary $\mathcal{V}$ using Kleene's closure. $\mathcal{K}$ is a string composed of information annotated in ▦ MATHDIAL , namely the MWP, step-by-step solution, and the students' solution attempt. We study locally-normalized models of the form

$$p_{\boldsymbol{\theta}}(u_{T+1} \mid u_1^T, \mathcal{K}, \mathcal{A}) = \prod_{n=1}^{N_{T+1}} p_{\boldsymbol{\theta}}([u_{T+1}]_n \mid [u_{T+1}]_1^{n-1}, u_1^T, \mathcal{K}, \mathcal{A}),$$

where $\boldsymbol{\theta}$ denotes the parameters of the model and $T$ is moved throughout the dialogue to evaluate each intermediate teacher turn. We either optimize these parameters by finetuning for 10 epochs or zero-shot prompting an LLM. When finetuning, we use an initial learning rate of $6.25e-5$ and linear learning rate decay without warm-up, and optimize the negative log-likelihood of the ground-truth response using the AdamW optimizer (Loshchilov and Hutter, 2019). We experiment with state-of-the-art pretrained Transformer (Vaswani et al., 2017) models and make use of the checkpoints provided by the transformers library (Wolf et al., 2020). In particular, we finetune BART (Lewis et al., 2020), Flan-T5 (Chung et al., 2022) which is based on T5 (Raffel et al., 2020) and was finetuned on the instruction-following flan collection (Longpre et al., 2023), as well as OPT (Zhang et al., 2022). Finally, we zero-shot prompt ChatGPT (Brown et al., 2020).

**Data split** We split our data into a training split containing $80\%$ of the conversations and a test set containing the remaining $20\%$. Around $60\%$ of the problems in the test set are also found in the training data, where at least one conversation was based on it, and therefore constitute our 'seen' split. The remaining $40\%$ are *unseen* during training and test the ability of the model to generalize to new problems. The dataset split is published with the dataset.

**Metrics** We assess our models using the sacrebleu (Post, 2018) implementation of BLEU (sBLEU) (Papineni et al., 2002), as well as BERTScore [3] (Zhang et al., 2020) between generated response ($u_{T+1}$) and annotated response ($\hat{u}_{T+1}$) for each teacher response in the conversation. Furthermore, in line with previous works (Dziri et al., 2022; Daheim et al., 2023), we report BERTScore and the token level F1 (KF1) between generated utterance and math word problem as a proxy for faithfulness. However, we note that an increase in these metrics can be caused by an increase in overlap, which may also indicate more telling and can be undesirable. However, finding good evaluation metrics for assessing the faithfulness of dialogue tutors remains an open problem. Finally, we measure the Uptake of the generated response (Demszky et al., 2021).

We propose two evaluation metrics for end-to-end tutoring, where a tutor model is evaluated interactively by using it to teach an LLM that simulates a student. Success@k measures the percentage of conversations where the student reaches the correct final answer at least once within the first $k$ turns (equivalent of % solve rate in prior work). Telling@k measures the percentage of conversations where the teacher explicitly tells the final answer before the student has reached it on their own within the first $k$ turns.

---

[3] We use the deberta-large-mnli checkpoint

| | sBLEU ($\uparrow$) | BERTScore($\uparrow$) |
|---|---|---|
| | | $(u_{T+1}, \hat{u}_{T+1})$ |
| Flan-T5$_{780M}$ | 8.0 | 53.0 |
| + question | 8.6 | 53.2 |
| + incorrect solution | 8.3 | 53.5 |
| + ground-truth | **9.5** | **55.0** |
| + all | 9.7 | 55.0 |

Table 4: Ablation on the influence of grounding information, which shows that the ground-truth solution gives the model the most valuable information.

## 6 Results

### 6.1 Tutor Response Generation

Table 3 shows our main results for the task of tutor response generation on 📊 MATHDIAL . A first general observation is that automatic metrics appear low when compared to state-of-the-art models on other dialogue data. This might be explained by two main challenges that tutoring models face: a high level of ambiguity when it comes to sound teaching strategies and complex problems that the models need be able to correctly assess. In contrast, the data that ground responses in other dialogue tasks often needs a lesser amount of interpretation.

Scaling models in terms of their parameter size is not directly reflected in improved metrics. This indicates that just using larger models might not be enough to build meaningful tutors on 📊 MATH-DIAL . Still, as shown in BERTScore and lexical overlap between response and grounding information, smaller models appear to rely more on the grounding information and might paraphrase less which might make teaching less engaging for students. Instruction tuning seems to have a largely positive effect in tutoring, as well. This is exhibited by the improvements that Flan-T5 yields over T5.

In order to be used in real-world settings, dialogue tutoring models need to be able to generalize to new problems. However, we find that there is still a large gap in the performance of all finetuned models between seen and unseen problems. This indicates a clear need to build models that can generalize better. Uptake on the other hand is generally high and for different models even higher than the ground-truth annotations. Finally, finetuned models tend to outperform zero-shot prompted GPT in terms of automatic metrics but the validity of them for evaluating such models may be questioned.

### 6.2 Influence of grounding information

📊 MATHDIAL provides a large set of annotations that can be used to ground the responses of dialogue tutors trained on it. Table 4 shows results obtained with Flan-T5$_{780M}$ when giving different information. The results show that the step-by-step solution is crucial for the model. Question and incorrect solution are not as crucial but are also often repeated by student or teacher throughout the dialogue. Future work can explore this information in more detail to improve tutoring models.

### 6.3 Human Evaluation

| Model | Coherence ($\uparrow$) | Correctness ($\uparrow$) | Equitable ($\uparrow$) |
|---|---|---|---|
| | 3-point | 0/1 | 3-point |
| Flan-T5$_{780M}$ | 2.85 | 0.89 | **2.19** |
| Flan-T5$_{3B}$ | 2.84 | **0.91** | 2.18 |
| OPT$_{1.3B}$ | 2.61 | 0.72 | 1.95 |
| ChatGPT | **2.89** | 0.43 | 1.43 |
| Ground-truth | 2.94 | 0.98 | 2.42 |

Table 5: Human evaluation shows that finetuning models on 📊 MATHDIAL increases their performance in terms of correctness and equitable tutoring.

Finally, we conduct a human evaluation according to three criteria: 1) **Coherence**: how coherent the teacher's response is with respect to the preceding dialogue, 2) **Correctness**: whether it is in itself correct, and 3) **Equitable tutoring**. Equitable tutoring describes how well the model provides the student with room for exploring the problem and solution space. We use three expert annotators that each annotate $n = 50$ responses. We obtain agreements of $\kappa = 0.29$, $\kappa = 0.69$, and $\kappa = 0.34$ for the three categories. We find that the ground-truth data that we have collected shows high scores in all three criteria which confirms its quality. Then, we find that small fine-tuned models perform much better in terms of correctness and equitable tutoring than a prompted large language model (ChatGPT), even though the latter is pretrained on much more data and has a significantly larger parameter count. This shows the importance of high-quality data for training meaningful tutors. The automatic metrics are only partially confirmed. For instance, Flan-T5$_{3B}$ is rated slightly better than Flan-T5$_{780M}$ in correctness despite lower automatic scores.

### 6.4 Interactive Evaluation of Dialogue Tutors

Good tutoring models need to maintain high quality not only when viewed per-utterance but especially over an entire conversation. In order to assess this, we use them to tutor an InstructGPT student and measure their success (Success@k), as well as the rate of telling (Telling@k). The tutor models are

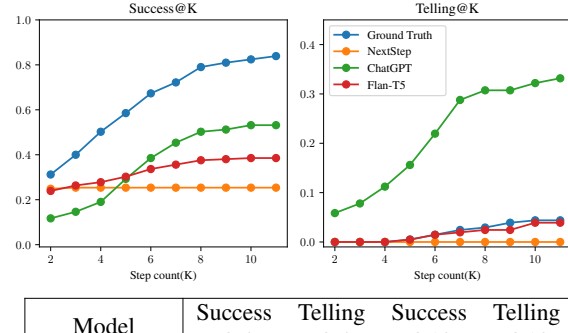

| Model | Success @5 | Telling @5 | Success @10 | Telling @10 |
|---|---|---|---|---|
| NextStep | 25% | 0% | 25% | 0% |
| ChatGPT | 29% | 16% | 53% | 32% |
| Flan-T5$_{780M}$ | 30% | <1% | 39% | 4% |
| Ground Truth | 59% | <1% | 82% | 4% |

Figure 5: Performance of our tutor model and 3 baselines on interactive tutoring of the student model. We find the model trained on 📊 MATHDIAL to have a similar success@5 rate with less telling.

used as outlined in the previous subsections and the student model uses the same settings as during data collection. We compare our Flan-T5$_{780M}$ model with a simple baseline that repeatedly asks "What is the next step?" (NEXTSTEP), ChatGPT, and the ground-truth conversations.

Figure 5 shows that NEXTSTEP has the lowest success rate, but never tells solutions by construction. ChatGPT, on the other hand, has a high success rate but also the highest rate of telling. This is a crucial shortcoming because high telling is counterproductive to effectively teach students. Flan-T5$_{780M}$ achieves a balance between the two and shows a similar amount of telling as the ground truth.

We note that the gap in success rate between Flan-T5$_{780M}$ and ChatGPT, at least in the initial steps, stems mostly from longer problems, as is evident from Figure 6. Overall, no model can match the success rate of the ground-truth annotations. This indicates a large room for future improvements and research.

## 7 Conclusion

We introduce a new framework for semi-synthetic dialogue dataset collection. We use it to collect a pedagogically rich dataset for tutoring math word problems that follow equitable tutoring practices and learning sciences research on scaffolding student understanding, called 📊 MATHDIAL. Our dataset consists of ca. 3k tutoring conversations grounded in math word problems from GSM8k.

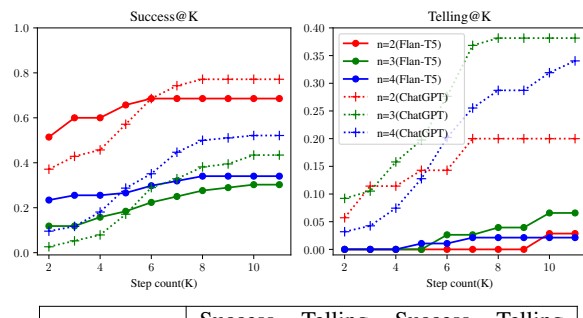

| Model | Success @5 | Telling @5 | Success @10 | Telling @10 |
|---|---|---|---|---|
| Problems with 2 step Solutions | | | | |
| Flan-T5$_{780M}$ | 65% | 0% | 68% | 2% |
| ChatGPT | 57% | 14% | 77% | 20% |
| Problems with 3 step Solutions | | | | |
| Flan-T5$_{780M}$ | 18% | 0% | 30% | 7% |
| ChatGPT | 17% | 19% | 43% | 38% |
| Problems with 4 step Solutions | | | | |
| Flan-T5$_{780M}$ | 27% | 1% | 34% | 2% |
| ChatGPT | 29% | 20% | 52% | 32% |

Figure 6: Performance of our tutor model and ChatGPT on interactive tutoring of the student model on problems with solutions of different lengths ($n$ is the number of steps in the ground truth solution). The performance of all models drops for problems with more than 2 step solutions.

We benchmark open-source models on the task of tutor response generation and show that smaller models finetuned on our 📊 MATHDIAL can significantly surpass the performance of much larger prompted LLMs. Moreover, in our proposed interactive tutoring simulation, the finetuned model achieves similar student-solving success as prompted LLM while keeping the direct telling rate lower. Nevertheless, models still require better reasoning over student solutions and better generalization to unseen problems.

Our dataset fills a crucial gap towards studying effective dialogue tutors at scale by providing a significantly larger amount of dialogues than other available corpora in one-on-one tutoring and provides a tough testbed towards better tutoring models. We hope that it can spark more research in this meaningful but understudied area of NLP.

## 8 Limitations

In this work, we used an LLM to simulate student confusion. However, we acknowledge that these models have a limited understanding of human learning and this is a key limitation in our dataset – certain kinds of student confusions may be under- or over-represented in our dataset. Future

work can focus on addressing this limitation.

Furthermore, in our setup, teachers were interacting with an LLM role-playing as a student. However, it is possible that some teachers might have learned to interact with the student model in a different way than they would do in the classroom. Moreover, it is also possible that some teachers may have lost motivation when found out they are not interacting with real students, leading to lower data quality. In the future, we would like to explore solutions to build better LLM-based student models (Zhou et al., 2023).

The methodology to collect the dataset was instantiated just for the domain of math reasoning. The collection of additional domain-specific datasets is necessary to further generalize the effectiveness of our methodology.

Inspired by previous work in scaffolding, we acknowledge our focus is on a subset of common teaching moves. However, this does not cover all the goals of human tutors, such as meta-cognitive support or building rapport with a student. Moreover, text tutoring limits teachers' use of additional instructional practices such as drawings.

Finally, measuring a student's immediate success in solving a problem does not capture all the aspects of student learning. From a learning perspective, focusing on and measuring long-term learning is desired. Therefore, even if students struggle to answer a specific problem correctly, teachers asking scaffolding questions requiring conceptual understanding offer even better promise for deeper, wider, and more long-term learning.

# 9 Acknowledgements

This project was made possible by an ETH AI Center Doctoral Fellowship to Jakub Macina with further support from the Asuera Stiftung and the ETH Zurich Foundation. Nico Daheim has received funding by the German Federal Ministry of Education and Research and the Hessian Ministry of Higher Education, Research, Science and the Arts within their joint support of the National Research Center for Applied Cybersecurity ATHENE. Mrinmaya Sachan acknowledges support from the Swiss National Science Foundation (Project No. 197155), a Responsible AI grant by the Haslerstiftung; and an ETH Grant (ETH-19 21-1).

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

## A  Dataset statistics

For NCTE, uptake is calculated on the teacher-student dialogue pairs while bigram entropy is calculated on all teacher utterances. For TalkMoves and TSCC, bigram entropy is calculated on all teacher utterances having more than three words, while uptake is calculated on teacher utterances immediately following student utterances if both have more than three words.

## B  Problem and confusion selection

While the problems in GSM8k are simple enough to be understood quickly by teachers, they remain challenging for students, who among others have to deal with equations or percentages. We follow the GSM8k reasoning format and prompt ChatGPT (gpt-3.5-turbo) with a 2-shot prompt. Given a prompt and a math word problem, we sample $n$ reasoning paths $r_i$ solutions from the model. We parse the first numerical answer $a_i$ after the model

generated "####" which represents the final result. Most of the generated outputs have this format and we discard all generations not following it. We sample $N = 50$ reasoning path candidates using the same settings as suggested by (Wang et al., 2023b). After sampling multiple reasoning pairs and corresponding answer pairs $(r_i, a_i)$ we use a majority vote over $a_i$ which does not lead to a ground truth answer $a$: $\arg\max_a \sum_{i=1}^n \mathbb{1}(a_i \neq a)$. We select problems with at most four solution steps. Since our initial experiments show the occurence of rounding errors, which related work finds to be more common in LLMs than humans (Frieder et al., 2023), we limit them by discarding confusions that are within $0.1$ of the original solution. Moreover, to filter out other simple calculation errors which are not interesting from a learning standpoint we parse all the intermediate equations which are in the format $<< a \times b = c >>$ and use a calculator to check for inconsistencies.

The full prompt used is:

```
Q: Natalia sold clips to 48 of her friends
in April, and then she sold half as many
clips in May. How many clips did Natalia
sell altogether in April and May?

A: Natalia sold 48/2 = «48/2=24»24 clips
in May. Natalia sold 48+24 = «48+24=72»72
clips altogether in April and May. ####
72

Q: Weng earns $12 an hour for babysitting.
Yesterday, she just did 50 minutes of
babysitting. How much did she earn?

A: Weng earns 12/60 = «12/60=0.2»0.2 per
minute. Working 50 minutes, she earned
0.2 x 50 = «0.2*50=10»10. #### 10
```

Of the problems in the GSM8k dataset, $5684$ problems were queried after eliminating problems with more than 5 steps in the solution. This yielded $2,313$ problems with at least one wrong solution. We then eliminated student solutions having fewer than $300$ characters (having too few characters makes it harder to pinpoint where exactly the error occurred) or more than $500$ characters (longer solutions require annotators to spend more time understanding the error), leaving us with $1,379$ wrong solutions. Finally, we eliminate problems where all $50$ or $49$ out of $50$ proposed solutions have the same (wrong) final answer, leaving us with our final set of 1131 problems.

# C  Student model

## C.1  Prompt

We use InstructGPT (text-davinci-003) with the following prompt using temperature sampling with $T = 0.4$ and no top-k truncation:

```
Student Persona: (STUDENT PERSONA)\n\n
Math problem: (MATH PROBLEM)\n\n
Student solution: (STUDENT SOLUTION)\n\n
Context: (STUDENT NAME) thinks their
answer is correct.  Only when the
teacher provides several good reasoning
questions, (STUDENT NAME) understands
the problem and corrects the solution.
(STUDENT NAME) can use a calculator and
thus makes no calculation errors.  Send
EOM tag at the end of the student
message.\n\n
(DIALOGUE HISTORY)
```

## C.2  Student characteristics

To build a dataset that would reflect students of various backgrounds, we use numerous student names associated with their given pronouns. List of all student characteristics based on prior work studying misconceptions in learning algebra (Booth et al., 2017):

- has a problem with understanding what steps or procedures are required to solve a problem.

- has a problem with understanding underlying ideas and principles and a recognition of when to apply them.

- struggle most with understanding what the problem is asking them to do.

- has difficulty determining which pieces of information are relevant and which are irrelevant to solving the problem.

- struggle to put the numbers in the correct order in the equation or determine the correct operation to use.

- struggle to recognize the problem type and therefore do not know what strategy to use to solve it.

## C.3 Common error cases

We manually screened some conversations and teacher feedback to understand common error cases of student model. The most common problem among them was the occurence of simple arithmetic errors (e.g. 7-2=9) and inconsistent student behaviour (e.g. student returning to the incorrect answer after figuring out the correct one in the previous utterance). These errors are captured in the teacher quality Likert scale rating of student behaviour. We acknowledge further analysis is needed to better understand the fine-grained student model behavior on problems with different numbers of steps e.g. by cognitive task analysis (Koedinger and McLaughlin, 2016).

## D Data collection interface

We use Prolific for data collection and hire annotators with teaching experience. To ensure the data quality we filter only annotators with 100% completion rate with more than 500 total submissions. All the payments to the annotators exceeded the US federal minimum wage and the final batch of annotators were paid the equivalent of $12/hour. The data collection interface is shown in Figure 7. Annotators were restricted to having a maximum of five conversations in one annotation session. One conversation takes ca. 6 minutes. Data collection took place over a period of 2 months.

### D.1 Annotation pipeline

For each annotator, we randomly assign a student and math word problem. Teachers were instructed to first analyze the student homework solution and then start the conversation to scaffold student problem understanding. Post-conversation questionnaire is filled out by teachers to rate the conversation and get feedback on the type of student error.

**Comparing solutions** As shown in Figure 8, the teacher first analyzes and compares the correct solution with the incorrect student solution (student confusion). The teacher marks the exact line of a first student error and categorizes the problem into the following categories:

- Reached correct solution but proceeded further

- Extra quantity or Missing quantity

- Unit conversion error

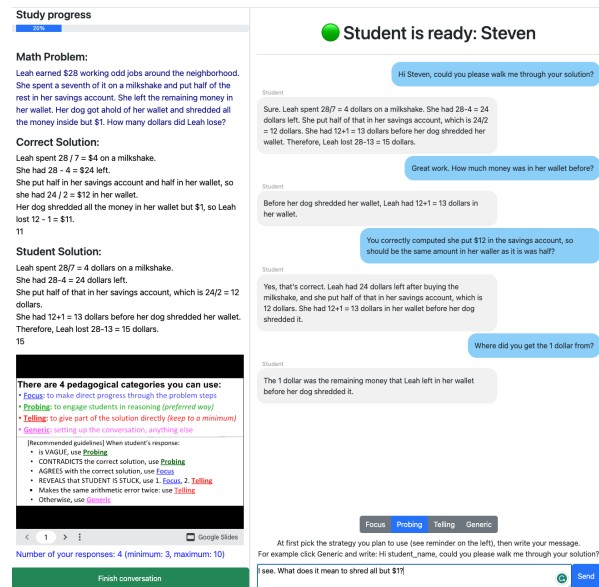

Figure 7: Web interface of the tool for collecting dialogue tutoring conversations. The left panel shows math word problem, correct solution, and student solution. The right panel contains conversation history, a panel for selecting the category of response, and a text area to send a response to the student. After clicking Send, the student model is immediately invoked using an internal API call.

- Calculation error easily solved by a calculator

- Missing / Wrong factual knowledge

- Misunderstanding of a question

- None of the above

**Tutoring conversation** Next, the teacher has a conversation (see Figure 7) with a student and uses scaffolding moves to help the student understand the problem. The conversation ends when the student correctly solves the problem or if the total conversation time exceeds 10 minutes.

**Post conversation questionnaire** Teacher fills the post conversation questionnaire as shown in Figure 9.

### D.2 Annotators training phase

We let annotators read best practices on how to have a productive conversation with students (cf. Section D.3 and D.4) and tested them on their understanding of our task afterwards. We started the data annotation with all the annotators able to successfully pass the test. Moreover, to improve the training phase we manually checked several conversations by each annotator in terms of the quality and usage of diverse scaffolding questions.

Figure 8: Teacher first compares student solution with the correct solution and marks the exact step of the error.

## D.3 Annotation Guidelines

Teachers were instructed to have a one-on-one tutoring session with different 6th-grade students. They were told that students received a math word problem for homework and submitted their solutions beforehand. In a tutoring conversation, teachers were asked to go through the student's solution and try to let the student understand using a series of sensemaking questions to support student reasoning and learning. Specifically, they were instructed to not just correct student solutions by telling what's correct/incorrect, but to give students the opportunity to explore the problem with a focus on core aspects, such as their chosen strategy. However, as the goal is to focus on conceptual errors, they were allowed to let students use calculators or correct their arithmetic mistakes.

## D.4 Teacher moves taxonomy

Table 2 refers to the details of teacher moves used during annotation. In summary, Focus comprises of all conversation elements that direct the student towards the solution without actually giving out any of the solution, while Probing attempts to develop reasoning skills and world knowledge relevant to the problem, but not necessarily specific to the given problem. Telling is giving out parts of the solution, either calculations or strategy or both. All other conversational elements, including trying to understand what the student has already tried, fall under Generic.

Most importantly, scaffolding questions that are productive for long-term learning are Focus and Probing. On the other hand, Telling represents

Figure 9: Post questionnaire.

giving out the partial or full answer to the student and should be mostly used when a student is stuck.

## D.5 Background for teacher moves

Scaffolding (Reiser, 2004; Anghileri, 2006) assists students to succeed in tasks that would otherwise be complex and differentiates between guidance (e.g. decomposing problem, clarifying) from cognitive activation (e.g. causing cognitive conflicts, activating prior knowledge (Limón, 2001)). The effective teacher moves to scaffold students' understanding have been studied extensively by analyzing and annotating real human tutoring conversations (Nye et al., 2014; VanLehn, 2011). Experienced teachers can through natural language guide students' focus and uncover misconceptions (Nye et al., 2014). The teacher moves in the form of scaffolding to support student understanding by asking open-ended questions, activating their prior knowledge, or causing cognitive conflicts (Limón, 2001). A teacher asking scaffolding questions provides learning opportunities for students to actively construct their knowledge. However, at the same time asking only difficult questions could lead to a loss of learner motivation and potentially the end of the dialogue. On the other hand, only constantly revealing answers does not lead to long-term learning.

### D.6 Postprocessing

As we are interested in real educational use cases for our tutoring system, we apply a safety filter to filter out conversations with any sensitive content. In particular, we use the Perspective API[4] to filter out conversations containing toxic content (<1%).

### D.7 Initial pilots

We initially explored two additional approaches of data collection: i) human-human conversations, and ii) synthetic generation by LLMs. The framework we used in the final data collection enables us to scalably create data since we are only reliant on one user who can quickly create entire conversations with the LLM, taking ca. 6 minutes per 7+ turn conversation. We found this more efficient and performant than both human-human conversations and synthetic data generation. Specifically, the human-to-human collection is too time-consuming (on average 15 minutes per conversation in our pilot experiments) and requires waiting times to synchronously connect participants (Choi et al., 2018), and synthetic generation has proven to be error-prone (see example in Figure 10); for example, models fail to understand student solutions and themselves make arithmetic errors that are not expected from teachers.

### E Interactive evaluation of tutoring

The student model in all 3 cases is an InstructGPT model (`text-davinci-003`) as defined in Section C.1, with the student name fixed to "Kayla". The first utterance of the teacher is hardcoded to "Hi Kayla, could you walk me through your solution?". For Flan-T5$_{780M}$ teacher model decoding, we used sampling without a beam search. For the ChatGPT teacher model (`gpt-3.5-turbo`), the following prompt is used:

```
A tutor and a student work together to
solve the following math word problem.\n
Math problem: (MATH PROBLEM)\n
The correct solution is as follows:
(CORRECT SOLUTION)\n
Your role is tutor.  The tutor is
a soft-spoken empathetic person who
dislikes giving out direct answers to
students and instead likes to answer
with other questions that would help
the student understand the concepts
```

---
[4]https://perspectiveapi.com

```
so students can solve the problem
themselves.
```

### F Human Evaluation Protocol

The following dimensions were rated by annotators:

- **Coherence** - *"The response naturally follows up on the previous utterance and context and has no logical conflicts with the context."*

- **Correctness** - *"The response is factually and mathematically correct and respects the learning concepts being taught."*

- **Equitable tutoring** - *"The response gives a learning opportunity for the student by providing space for reflection, explanation, pointing to follow-up challenge, or engaging the student in other ways.*

We use a 3-point Likert scale ranging from 1 (poor) and 3 (very good) for coherence and equitable tutoring and a binary scale for correctness.

ChatGPT prompt is the same as in the interactive tutoring scenario (Section E) with an additional section containing student solution.

| Context | Conversation |
| --- | --- |
| *Q*: Sam memorized six more digits of pi than Carlos memorized. Mina memorized six times as many digits of pi as Carlos memorized. If Mina memorized 24 digits of pi, how many digits did Sam memorize?

*Ground truth*: Carlos memorized 24/6=4 digits of pi. Sam memorized 4+6=10 digits of pi. Final Answer: 10

*Student solution*: Step 1: Carlos memorized 18 digits of pi. Step 2: Mina memorized 24 digits of pi. Step 3: Sam memorized 6 more digits of pi than Carlos, so he memorized 24 digits of pi (18 + 6 = 24). Final Answer: 24 | T: (probing) You outlined that Carlos memorized 18 digits of pi - where did you get this figure from?

S: I got this figure by taking the number of digits that Mina memorized (24) and subtracting the number of digits that Carlos memorized more than Mina (6). So 24 - 6 = 18.

T: (probing) If the question says that Mina memorised six times as many digits of pi as Carlos memorised, and she memorised 24, what do you think that means for the digits of pi that Carlow memorised?

S: It means that Carlos memorized 4 digits of pi (24/6 = 4).

T: (focus) That's brilliant - really well done. So, if that means that Carlos memorised 4 digits of pi, and the question says that Sam memorised six more digits of pi than Carlos, how many digits of pi has Sam memorised?

S: Sam has memorized 10 digits of pi (4 + 6 = 10).

T: (focus) That's absolutely perfect! So, based on that, how many digits of pi did Carlos, Mina and Sam each memorise?

S: Carlos memorized 4 digits of pi, Mina memorized 24 digits of pi, and Sam memorized 10 digits of pi.

T: (focus) Well done Luca! You got it right! |

Table 6: Examples of 📑 MATHDIAL conversations. *T* refers to a teacher utterance, *S* refers to a student utterance. Each conversation is grounded in the correct solution and student solution. Bold text is information for the reader indicating error categories.

```
Prompt:
A proficient educator assists a learner by providing just a few clues to help him/her in the correct direction. The
student should take the lead in reasoning and computations. Generate a tutoring conversation between a math teacher and
an 8th-grade student Leslie unable to solve the following math word problem correctly.

Math Problem: Jeremy decides to take Tylenol because of the pain he is experiencing. He decides to take 1000 mg every 6
hours for 2 weeks.  How many pills do they take if they are each 500 mg?

Correct solution: He takes 1000/500=<<1000/500=2>>2 pills each time. He takes them 24/6=<<24/6=4>>4 times a day. So he
takes 4*2=<<4*2=8>>8 pills per day. He takes pills for 2*7=<<2*7=14>>14 days. So he takes 8*14=<<8*14=112>>112 pills

InstructGPT:
Teacher: Hi Leslie, it looks like you're having some trouble with this math word problem. Can you tell me what you're
struggling with?

Leslie: Yeah, I'm not sure how to figure out how many pills Jeremy needs to take.

Teacher: Alright, let's break down the problem and see if we can solve it step by step. First, let's think about how much
Tylenol Jeremy needs to take each day. He's taking 1000 mg every 6 hours, so how many times does he need to take it each
day?

Leslie: He needs to take it 3 times a day because 24 / 6 = 3.

Teacher: That's correct! Now let's figure out how much Tylenol he takes each day in total. If he takes 1000 mg each time,
and he takes it 3 times a day, how much does he take in total each day?

Leslie: He takes 3000 mg each day because 1000 x 3 = 3000.

Teacher: Excellent! Now let's figure out how many pills he needs to take each day. Each pill is 500 mg, so how many pills
does he need to take each day to get his 3000 mg?

Leslie: He needs to take 6 pills each day because 3000 / 500 = 6.

Teacher: That's correct! Now let's think about how many pills he needs to take in total over the 2 weeks. If he needs to
take 6 pills each day, and there are 7 days in a week, how many pills does he need to take in total over 2 weeks?

Leslie: He needs to take 84 pills in total because 6 x 7 x 2 = 84.

Teacher: Great job, Leslie! You've done a fantastic job breaking down the problem and solving it step by step. Well done!
```

Figure 10: In our initial pilot study we observed that synthetic data generation by InstructGPT strictly followed the same structure of only asking next-step questions (highlighted in yellow) and was prone to inconsistencies in factual correctness and order of steps (highlighted in red).