# OpenReview forum: "MathDial: A Dialogue Tutoring Dataset with Rich Pedagogical Properties Grounded in Math Reasoning Problems"
_EMNLP/2023/Conference — EMNLP 2023 Findings_

### Official Review · Reviewer_q15M · 2023-08-05

**Typos Grammar Style And Presentation Improvements:** N/A
**Soundness:** 3

**Excitement:**

5: Transformative: This paper is likely to change its subfield or computational linguistics broadly. It should be considered for a best paper award. This paper changes the current understanding of some phenomenon, shows a widely held practice to be erroneous in someway, enables a promising direction of research for a (broad or narrow) topic, or creates an exciting new technique.

**Missing References:**

N/A

**Paper Topic And Main Contributions:**

# Summary

This is a dataset paper that discusses the step-wise reasoning to solving math problems. These steps are generated in textual formats using models such InstructGPT and Flan-T5 (750M). The dataset contains 3k tutoring conversations. The purpose of this work is to understand the performance of GPT models as tutors instead of problem solvers, because of their tendency to solve the problem before tutoring a student. The goal is to analyze the generated dataset to build faithful, verifiable and equitable systems.

**Questions For The Authors:**

1. Why is the construction of a dataset more powerful discussion on this topic, rather a model based comparison approach? Is it to set a ground truth that is different from existing WoZ datasets? How do you manage to show this difference in the current work?

**Reasons To Accept:**

## Reasons
1. The work explores a more cognitive aspect (with relevant citations) to the use of dialogue in tutoring systems. This allows to discuss the idea that there are aspects to learning for the student from working out a problem step-wise. Using this dataset, it is possible to analyze and produce better models fine-tuned for this task.

2. Unlike the standard user study based approach, this work focuses on analyzing the closeness to human tutoring with GPT models. This could have pedagogical implications with the use of LLMs in Education.

## Contributions

1. A dataset for further analysis / discussions with the rise and use of LLMS  in Education

2. Promoting the use of AI agent instructors who can assist human instructors in helping students with a learning task, using grounding cognitive theories and applied cognitive theories in the use of dialogue based tutoring systems.

**Reasons To Reject:**

## Reasons / Weaknesses

I don't find strong reasons to reject, however a strong potential exploration which could be added to this paper is the idea of cognitive task analysis in Math

Koedinger, Kenneth R., and Atsushi Terao. "A cognitive task analysis of using pictures to support pre-algebraic reasoning." Proceedings of the twenty-fourth annual conference of the Cognitive Science Society. Routledge, 2019.

Koedinger, Kenneth R., and Elizabeth A. McLaughlin. "Closing the Loop with Quantitative Cognitive Task Analysis." International Educational Data Mining Society (2016).



**Reproducibility:**

4: Could mostly reproduce the results, but there may be some variation because of sample variance or minor variations in their interpretation of the protocol or method.

**Reviewer Confidence:**

3: Pretty sure, but there's a chance I missed something. Although I have a good feel for this area in general, I did not carefully check the paper's details, e.g., the math, experimental design, or novelty.

---

> ### Author Rebuttal · Authors · 2023-08-28
>
> We thank the reviewer for their insightful comments and references. We appreciate the reviewer *highlighting that our work could highly impact the use of LLMs in education*. Cognitive task analysis is an interesting perspective and our dataset could be potentially (with smaller additional annotation of knowledge components) used to study how teachers break down math problems into smaller subproblems and guide students. We will add these sources to section 2.2 and use quantitative cognitive task analysis in future work.
>
> **Why is the construction of a dataset more powerful discussion on this topic, rather a model based comparison approach?** We decided to focus on the data collection methodology because there is a lack of datasets available in the dialogue tutoring domain. Data is the most important asset in this domain and without it no experimentation could be done. Our work opens a new avenue of modeling approaches for the understudied domain of educational tutoring, and allows for building new models on a larger scale (due to the significantly larger size than previous corpora) and extensive grounding annotations.

---

### Official Review · Reviewer_i8Bf · 2023-08-05

**Soundness:** 4

**Excitement:**

4: Strong: This paper deepens the understanding of some phenomenon or lowers the barriers to an existing research direction.

**Paper Topic And Main Contributions:**

This paper produces a novel dataset of 3k math tutoring conversations, collected by having teachers interact with a LLM.  Once the data is collected, the authors benchmark several baseline approaches, finding fine-tuning approaches perform better.  The authors also have the student LLM interact with the trained tutoring models, showing a gap between current methods and human tutors.


**Questions For The Authors:**

Did you have multiple teachers judge LLM student interactions in Figure 3?  Do you have an IAA metric to report?

389: “88.29% of the time they were able to do so” <- when the confusion was not able to be resolved, how did this interact with the teacher judgments on the simulation ability in Figure 3?


**Reasons To Accept:**

Authors propose a novel framework to collect tutoring data by capturing common LLM errors, which capture student misconceptions in the majority of cases.  A LLM to interact with a tutor conditioned on the error.  This approach incorporates past educational work on pedagogical tutoring strategy.

This dataset is an exciting contribution to the dialogue tutoring community and could encourage more work in this area.

Analysis of different models trained to produce tutoring responses is comprehensive and insightful.  End-to-end evaluation with a trained tutor model and the LLM student is an interesting finding and shows there is value to exploring this task as a community.

**Reasons To Reject:**

While the authors conduct human evaluation, there is low-to-moderate agreement among annotators for multiple tasks.  I wonder if more training of annotators or a more specific annotation task might have mediated some of these issues and improved the strength of these results.


**Reproducibility:**

4: Could mostly reproduce the results, but there may be some variation because of sample variance or minor variations in their interpretation of the protocol or method.

**Reviewer Confidence:**

4: Quite sure. I tried to check the important points carefully. It's unlikely, though conceivable, that I missed something that should affect my ratings.

---

> ### Author Rebuttal · Authors · 2023-08-28
>
> We thank the reviewer for their insightful comments. We are encouraged that the reviewer appreciates *the value of exploring this task as a community, incorporating past educational work, and thorough analysis and evaluation*. Please find our responses to the points raised below:
>
> **Annotators agreement**: Regarding section 4.2, a moderate agreement between human annotation of probing vs. focus has been observed in previous works. It reflects the mental state of the tutor and the effect they expected to achieve with the utterance. Since the post hoc annotators do not have access to said mental state, the agreement is expected to be low.\
> Regarding section 7.3, we agree both better training of annotators and refining the specificity of the annotation task might have mediated some of these issues and improved the robustness of these findings.
>
> **Did you have multiple teachers judge LLM student interactions in Figure 3?** We had every teacher self-report whether or not they were able to resolve the confusion of the student immediately after the completion of the conversation, hence we have one annotation per conversation. The same math problem and confusion may have been encountered by multiple teachers (55.7% of math problems have been annotated by more than 3 annotators), but since the student and teacher behaviors are different for each conversation, we cannot compare annotations across this.
>
> **When the confusion was not able to be resolved, how did this interact with the teacher judgments on the simulation ability in Figure 3?** We thank the reviewer for suggesting this interesting analysis. For conversations that the teachers could successfully resolve, the mean of the teacher judgment of student behavior is 3.59, while it is only 2.34 when the conversation was not resolved. The difference is even more stark for the teacher judgment of student confusion, where successful conversations had a mean rating of 3.54 but 1 on unresolved conversations. This seems to indicate that teachers felt that all confusions that could not be resolved were unrealistic, while student behavior was also less realistic than the ones they were able to resolve. We shall add this analysis to the camera-ready version.

---

### Official Review · Reviewer_jaAZ · 2023-08-12

**Soundness:** 3

**Excitement:**

3: Ambivalent: It has merits (e.g., it reports state-of-the-art results, the idea is nice), but there are key weaknesses (e.g., it describes incremental work), and it can significantly benefit from another round of revision. However, I won't object to accepting it if my co-reviewers champion it.

**Paper Topic And Main Contributions:**

The main contributions of this paper are the introduction of a novel framework for the creation of semi-synthetic dialogue datasets, wherein genuine experts collaborate with LLMs to emulate plausible students. Furthermore, a comprehensive tutoring dataset consisting of 3,000 dialogues is built based on the framework. This dataset is augmented with extensive grounding annotations. The experimental results are effective and extensive.

**Questions For The Authors:**

1. Why there are three settings  (MATHDIAL, seen, and unseen) in Table 3?
2. In Appendix E.1, the model is finetuned given the teacher's move as one of the inputs. However, is it should be decided by the model itself?
3. Other questions in reasons to reject.

**Reasons To Accept:**

1. The motivation is very straightforward and the targeted problem is very valuable.
2. The dataset can fill a crucial gap in studying effective dialogue tutors at scale by providing a significantly larger amount of dialogues than other available corpora in one-to-one tutoring.

**Reasons To Reject:**

1. The paper is not well-organized and hard to follow. For example, Table 1 is presented on Page 3 and introduced on Page 5. Also, Figure 2 does not bring any understanding improvements to the paper.
2. The collection methodology relies on using LLMs to simulate students, which is crucial that the model can do so realistically to create high-quality data as claimed in Section 4.1. However, according to Figure 3, it seems can not lead to the conclusion that LLMs can simulate students since the percentage gap between red and blue is relatively small. In addition, as LLMs are tuned to follow instructions, a real human student may not strictly follow the instructions and there are other problems that may occur in practice.
3. The experimental details are not clear. For example, the details of input and output during finetuning and the loss functions, also the inference details.
4. The evaluation is not suitable or not clear. According to Lines 439-454, the metrics are evaluated between generated response and annotated response at the end of the dialogue session. However, intermediate tutoring processing is also very important in this situation, and sometimes even outweighs the final answer.

**Reproducibility:**

3: Could reproduce the results with some difficulty. The settings of parameters are underspecified or subjectively determined; the training/evaluation data are not widely available.

**Reviewer Confidence:**

4: Quite sure. I tried to check the important points carefully. It's unlikely, though conceivable, that I missed something that should affect my ratings.

---

> ### Author Rebuttal · Authors · 2023-08-28
>
> We thank the reviewer for their comments. We are happy the reviewer shares the *motivation to address this important and valuable problem of studying effective dialog tutors at scale*. We apologize for any difficulty the reviewer faced due to the placement of figures and will improve their placements to make the camera-ready version easier to understand. We try our best to respond to their queries hereafter:
>
> **Figure 2 Relevance**: Figure 2 aids the reader by providing a quick overview of our data collection methodology.
>
> **Figure 3 Meaning**:  We do not focus on the gap between the red and blue bars for our conclusions. Replacing students with LLMs leads to 2 possible sources of non-realism in our model: first, the student confusion may be unrealistic, and second, the behavior of the student while the teacher explains their error to them might be unrealistic. The red and blue bars represent how well these two have been mitigated respectively. The best case scenario, therefore, would be to have the entire weight on 5 for both bars. \
> Based on the data in Figure 3, we can see that the distribution is skewed towards the highest Likert rating of 5 (with an average of 3.45 and a median of 4 for both measures), indicating that the majority of simulations are assessed to be realistic by the expert annotators. To put our message across in a better manner, we will complement the Likert distribution figure with corresponding summary statistics in the camera ready.
>
> **Evaluation clarifications**: We study the continuation of the dialogue $u_{t+1}$ given a preceding set of turns $u_{1:t}$ and follow standard dialogue evaluation protocols used in [1], where $t$ is moved throughout the conversation, such that we evaluate each teacher turn in the dialogue. Therefore, evaluation contains intermediate situations. We train our models to predict the next response using cross-entropy to the ground-truth response. During inference, we use beam search with 4 beams. Thanks a lot for pointing this out, we will clarify the notation used in Section 6 and Appendix E.1 in our next version.
>
> **Why there are three settings (MATHDIAL, seen, and unseen) in Table 3?** To compare the generalizability of models to unseen math problems. This is a first step to studying the generalizability of tutoring models before evaluating on other subjects. As a reminder, our dataset consists of 1131 math problems. The test set contains around 60% of seen problems, where the model has seen at least one conversation on it during training, and 40% of unseen problems, which the model has not encountered. Our results in Section 7.1 indicate that generalization to unseen math problems is one of the open problems for the community to address that is enabled by our collected dataset. The setup is described in further detail in L434-438.
>
> **In Appendix E.1, the model is finetuned given the teacher's move as one of the inputs. However, is it should be decided by the model itself?** Yes, that's indeed an important variant of evaluation more closely following real-world scenarios as indicated by [1]. Our dataset allows this evaluation scenario and we will be happy to add this experiment to Table 3 in the camera ready. Most importantly, we will encourage the community to follow it.
>
> [1] Jakub Macina, Nico Daheim, Lingzhi Wang, Tanmay Sinha, Manu Kapur, Iryna Gurevych, and Mrinmaya Sachan. 2023. Opportunities and Challenges in Neural Dialog Tutoring. In Proceedings of the 17th Conference of the European Chapter of the Association for Computational Linguistics, pages 2357–2372, Dubrovnik, Croatia. Association for Computational Linguistics.

---

### Official Review · Reviewer_pp3K · 2023-08-15

**Soundness:** 3

**Excitement:**

3: Ambivalent: It has merits (e.g., it reports state-of-the-art results, the idea is nice), but there are key weaknesses (e.g., it describes incremental work), and it can significantly benefit from another round of revision. However, I won't object to accepting it if my co-reviewers champion it.

**Paper Topic And Main Contributions:**

This paper proposed a dataset of 3k one-to-one teacher-student tutoring dialogues grounded in multi-step math reasoning problems. The dataset is collected under the WOZ framework, where the human teachers are paired with LLM students to generate dialogues. The dataset also defines different teacher moves and student characteristics, which are valuable for further analysis and larger scale data collection. The authors conduct detailed analysis and evaluation on the dataset to reveal the challenges of LLMs in tutoring students.

**Questions For The Authors:**

I wonder how much “teaching skill” can LLM learn from this dataset. After being trained on this dataset, can the model learn to educate students in other domains? For example, can the LLM use the teacher's moves to help a student with physics problems?

**Reasons To Accept:**

1.	The proposed data collection framework can effectively collect dialogues and is reproducible.
2.	Detailed and insightful analysis on the collected dataset.
3.	The collected dataset is potentially useful for developing a LLM teacher.

**Reasons To Reject:**

The major disadvantages have been discussed in the limitations section. One concern is that the “student” played by LLM might be more active than real students, and the humans teachers might not be able to maintain a high teaching quality during the data collecting procedure, which could bring more challenges to apply this dataset in real-world applications.

**Reproducibility:**

4: Could mostly reproduce the results, but there may be some variation because of sample variance or minor variations in their interpretation of the protocol or method.

**Reviewer Confidence:**

4: Quite sure. I tried to check the important points carefully. It's unlikely, though conceivable, that I missed something that should affect my ratings.

---

> ### Author Rebuttal · Authors · 2023-08-28
>
> We thank the reviewer for their comments. We are encouraged by the reviewer valuing the *data collection framework, teacher moves, analyses, and usefulness of the dataset for developing better LLM tutors*. We try our best to respond to their queries hereafter:
>
> **Responsiveness of InstructGPT Student**: We acknowledge that we cannot simulate real students perfectly, but we have incorporated several methods to mitigate this problem and we will highlight them better in the paper.\
> Firstly, we prime the student model with a prompt encouraging them to stick to their original (incorrect) answer unless convincing guidance has been provided by the teacher.\
> Secondly, we would like to reiterate that the dataset is intended to model teacher behavior in diverse teaching situations.\
> Finally, we asked teachers to rate conversations with students, and the majority of the conversations were rated rather positively as shown in Figure 3.\
> We are releasing the teacher ratings of how realistic the student confusion and student behaviors are, making it possible to train only on more realistic cases by filtering out those with low scores, should one choose this.
>
> **Transfer to Other Subjects**: In this work, we study the transfer to unseen math word problems and point out (in Section 7.1) an already significant gap in the performance. The teacher moves in our study and decomposition to the subproblems could be applied across a variety of learning subjects, given a set of problems. However, transfer to other subjects remains an open research problem and we believe our dataset could enable more research in this direction. We plan to analyze and address the transfer to other subjects in future work.\
> We would like to emphasize that our dataset is the first to allow studying this problem at such a scale.

---

### Meta-Review · Area_Chair_rdZg · 2023-09-17

**Recommendation:** 4

**Metareview:**

This paper introduces a novel dataset, MathDial, featuring 3k tutoring dialogues centered around multi-step math reasoning problems. The dataset was collected using a framework where real teachers interact with a Large Language Model (LLM) scaffolded to represent common student errors. The authors propose a taxonomy of teacher moves that promote student learning opportunities, and demonstrate the potential of MathDial to fine-tune language models into more effective tutors. The paper is submitted for consideration in the Resources and Evaluation and Dialogue and Interactive Systems tracks.

Reviewers largely agreed on the significant value of the proposed dataset to the community, particularly in the context of dialogue-based tutoring systems. They appreciated the paper's novelty, the potential impact of the work, and the depth of the analysis provided. Several reviewers pointed out that the dataset can fill a crucial gap in studying effective dialogue tutors at scale, and it could encourage more work in this area. However, the reviewers also raised several concerns and questions which were tried to address in rebuttal. These include the role of the LLM as a student, the realism and quality of the collected dialogues, and the clarity of the experimental details. The consensus among the reviewers was that while being an interesting field of study, the paper could significantly benefit from addressing these concerns and questions in the final version. The authors are strongly encouraged to take these comments into consideration to further improve their paper.

Overall, this submission is seen as an important contribution to the field of dialogue tutoring and has the potential to stimulate further research in this area. In addition, I encourage the authors to address the reviewers' feedback to further strengthen the paper.

---

### Decision · Program_Chairs · 2023-10-07

**Decision:**

Accept-Findings

**Comment:**

This paper introduces a novel dataset, MathDial, featuring 3k tutoring dialogues centered around multi-step math reasoning problems. The dataset was collected using a framework where real teachers interact with a Large Language Model (LLM) scaffolded to represent common student errors. The authors propose a taxonomy of teacher moves that promote student learning opportunities, and demonstrate the potential of MathDial to fine-tune language models into more effective tutors. The paper is submitted for consideration in the Resources and Evaluation and Dialogue and Interactive Systems tracks.

Reviewers largely agreed on the significant value of the proposed dataset to the community, particularly in the context of dialogue-based tutoring systems. They appreciated the paper's novelty, the potential impact of the work, and the depth of the analysis provided. Several reviewers pointed out that the dataset can fill a crucial gap in studying effective dialogue tutors at scale, and it could encourage more work in this area. However, the reviewers also raised several concerns and questions which were tried to address in rebuttal. These include the role of the LLM as a student, the realism and quality of the collected dialogues, and the clarity of the experimental details. The consensus among the reviewers was that while being an interesting field of study, the paper could significantly benefit from addressing these concerns and questions in the final version. The authors are strongly encouraged to take these comments into consideration to further improve their paper.

Overall, this submission is seen as an important contribution to the field of dialogue tutoring and has the potential to stimulate further research in this area. In addition, I encourage the authors to address the reviewers' feedback to further strengthen the paper.